# Common genetic variation associated with Mendelian disease severity revealed through cryptic phenotype analysis

David R. Blair [1✉], Thomas J. Hoffmann [2,3] & Joseph T. Shieh [1,2✉]

Clinical heterogeneity is common in Mendelian disease, but small sample sizes make it difficult to identify specific contributing factors. However, if a disease represents the severely affected extreme of a spectrum of phenotypic variation, then modifier effects may be apparent within a larger subset of the population. Analyses that take advantage of this full spectrum could have substantially increased power. To test this, we developed cryptic phenotype analysis, a model-based approach that infers quantitative traits that capture disease-related phenotypic variability using qualitative symptom data. By applying this approach to 50 Mendelian diseases in two cohorts, we identify traits that reliably quantify disease severity. We then conduct genome-wide association analyses for five of the inferred cryptic phenotypes, uncovering common variation that is predictive of Mendelian disease-related diagnoses and outcomes. Overall, this study highlights the utility of computationally-derived phenotypes and biobank-scale cohorts for investigating the complex genetic architecture of Mendelian diseases.

[1] Division of Medical Genetics, Department of Pediatrics, Benioff Children's Hospital, San Francisco, CA, USA. [2] Institute for Human Genetics, San Francisco, CA, USA. [3] Department of Epidemiology and Biostatistics, University of California, San Francisco, CA, USA. ✉email: David.Blair@ucsf.edu; Joseph.Shieh2@ucsf.edu

Advances in sequencing technology, cohort generation, and data dissemination have enabled the rapid identification of thousands of rare genetic variants associated with Mendelian diseases[1,2]. A great deal of this success can be attributed to their relatively simple genetic architectures; Mendelian diseases are predominantly caused by deleterious alleles clustered within a limited number of genomic loci. Nevertheless, clinical heterogeneity is commonly observed among diagnosed cases[1,3,4]. For example, Marfan Syndrome, an autosomal dominant disorder caused by mutations in the *FBN1* gene, is associated with cardiovascular, ocular, skeletal and even pulmonary abnormalities. Individuals with pathogenic *FBN1* alleles rarely manifest all of the associated symptoms[5], and even individuals within the same family can display disparate phenotypes[6]. Some of the clinical variability observed among Mendelian disease cases is attributable to allelic heterogeneity[1,3], but multiple lines of evidence also suggest a role for environmental and genetic background effects[4,7–11].

The identification of specific factors that modify Mendelian disease severity is inherently limited by the low prevalence of these disorders. Generally, it is difficult (but not impossible[12,13]) to construct cohorts of affected cases that are large enough to identify genetic and environmental modifiers, especially if they have relatively modest effect sizes. Given this limitation, many studies that investigate modifier effects have relied on model organisms[14,15] or the integration of orthogonal analyses[16,17]. As an alternative approach, we and others hypothesize that some Mendelian disorders may represent the severely affected extreme of a spectrum of pathologic variation. For conditions like familial hypercholesterolemia[18], hereditary breast cancer[19], and long QT syndrome[20], this relationship is well documented, and large biobank datasets have recently enabled investigators to examine the interplay between rare pathogenic variation and common polymorphisms[21,22]. In these examples, however, the analyses were possible because the condition of interest mapped to a univariate, quantitative phenotype. For Mendelian disorders that instead map to high-dimensional arrays of disparate symptoms, investigating the interplay between common and rare genetic variation becomes substantially more difficult.

In this work, we describe a probabilistic, model-based approach that infers latent quantitative traits that capture Mendelian disease severity using their diagnosed symptoms (cryptic phenotype analysis). We then systematically test the method on 50 different Mendelian disorders in two independent patient cohorts (UCSF Clinical Data Warehouse [UCSF], UK Biobank [UKBB][23]), uncovering multiple traits that reliably summarize disease severity. To validate these results, we use exome-sequencing data to demonstrate that pathogenic variation in known disease genes is associated with the inferred traits. Finally, we perform genome-wide association studies (GWAS) to identify common variation (summarized using polygenic scores; PGS) that is associated with cryptic phenotype severity and Mendelian-disease-related outcomes. This approach replicates the known architecture of a well-characterized genetic condition (α-1-antitrypsin deficiency [A1ATD]) while also identifying common variant modifiers for two Mendelian kidney diseases: Alport syndrome (AS) and autosomal dominant polycystic kidney disease (ADPKD). Overall, our study suggests that phenotype-driven approaches applied to biobank-scale data represent a powerful method for investigating the complex genetic architecture of rare diseases.

assumes that the Mendelian disorder of interest maps to the severely affected extreme of a spectrum of phenotypic variation (Fig. 1a, upper left). This implies that disease-related morbidity is not limited to the Mendelian cases but is instead spread throughout a larger subset of the population. Critically, this spectrum of variation cannot be measured directly. Instead, the trait is analyzed implicitly by a clinician, who translates their observations into a set of symptoms (Fig. 1a, lower left). These symptoms are then documented in the medical record, typically as a combination of structured and unstructured data. Building upon previous work[9,24], we aligned structured electronic medical record (EMR) data (i.e. ICD10 diagnostic codes[25], see Supplementary Fig. 1 and the "Methods" section) to the symptoms annotated within the human phenotype ontology[26]. This enabled us to construct a symptom matrix that encodes the severity of specific Mendelian diseases (Fig. 1a, right). This symptom matrix can then be used to recover cryptic, quantitative traits that summarize disease variability (Fig. 1b).

The process of decoding an observed symptom matrix into an underlying cryptic phenotype is equivalent to a form of matrix decomposition (Fig. 1b). In this scenario, the symptom matrix is decomposed into a risk function (Fig. 1b, upper right) and collection of one or more latent phenotypes (Fig. 1b, lower right). There are numerous ways to perform matrix decomposition. Using methods developed for machine learning[27,28], we designed a simple probability model for the observed symptom matrix that preserved its binary nature and enabled accurate, scalable inference of the desired latent phenotypes (see rge "Methods" section). Note, the recovery of these phenotypes is inherently limited by the loss of information that occurs when translating quantitative traits into binary symptoms. Therefore, inferred cryptic phenotypes will be inherently noisy (see Fig. 1b, lower right for example) unless the matrix contains hundreds of distinct symptoms, which is unrealistic for most diseases.

There is no guarantee that cryptic phenotypes inferred using this approach capture the severity of the intended Mendelian diseases, as the method is unsupervised. Therefore, we performed multiple analyses to ensure that the inferred traits reliably captured the phenotypic variability of interest (see Fig. 1c, study overview). We hypothesized that genetic factors associated with this variability are consistent across the full spectrum of phenotype severity. As a result, genetic modifiers identified in more mildly affected individuals should be predictive of outcomes in Mendelian disease cases. To test this, we used GWAS to identify common variation associated with each cryptic phenotype (Fig. 1c, bottom). Using a withheld sample of unrelated control and rare-disease affected individuals (as determined by exome data), we confirmed that the identified common variant effects were indeed associated with cryptic phenotype severity, disease-related laboratory measurements, and symptom onset/progression.

## Results

### A phenotype-driven approach to identifying Mendelian disease modifiers. Figure 1 outlines the approach taken to identify common-variant modifiers of Mendelian disease severity. It

**Quantifying disease severity using cryptic phenotype analysis (CPA).** Cryptic phenotype inference relies on fitting a generative probability model to observed symptom matrices. Due to the unsupervised nature of this inference, the latent phenotypes inferred by this approach are not guaranteed to capture the severity of the desired Mendelian disease. To circumvent this issue, we performed cryptic phenotype inference only for those diseases that: (1) mapped to specific diagnoses available in structured EMR data and (2) had prevalence of at least $10^{-5}$ in the UCSF dataset (to ensure adequate sample size for validation, see Supplementary Data 1 for complete list). Generative probability models were fit to the symptom matrices for each of the 50 Mendelian disorders meeting these criteria within both the UCSF

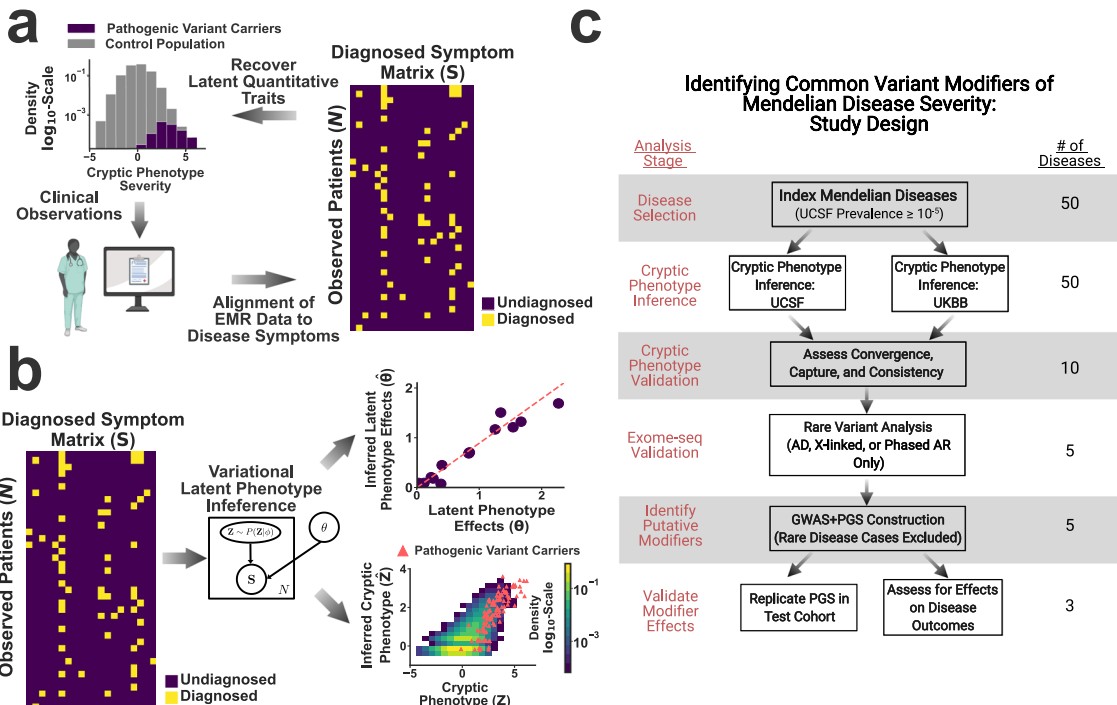

**Fig. 1 A phenotype-driven approach to identifying common variant modifiers. a** Schematic illustrating the assumptions underlying cryptic phenotypes and the proposed workflow. **b** Illustration of the model-based approach to symptom matrix decomposition and cryptic phenotype recovery. **c** Flow diagram describing the approach to inferring and validating cryptic phenotypes, which were subsequently used to identify common variant modifiers. UCSF: UCSF Clinical Data Warehouse; UKBB: UK Biobank. **a** and **c** were created using Biorender.com.

($N > 1.2$ million) and UKBB ($N > 500,000$) datasets. Consistent models were recovered for 38 of the 50 disorders (see the "Methods" section), with the remainder suffering from convergence issues in at least one of the two datasets (Fig. 1c).

To ensure that the inferred cryptic phenotypes captured the variability of the intended Mendelian disease, we assessed whether the trait was systematically elevated among withheld, diagnosed cases (Fig. 2a, exemplar Mendelian disease HHT). For 31 of the 38 disorders, the cryptic phenotypes were significantly increased among the diagnosed cases in the UCSF dataset (Bonferroni-corrected bootstrapped $P$-value $< 0.05$, Supplementary Data 4). To verify that the cryptic phenotypes were not dataset dependent, symptom matrix probability models were independently inferred using the UKBB, a population with different ascertainment, demographics, and healthcare infrastructure[23]. For 18 of the 31 disorders, the model inferred within the UKBB reproduced the elevated cryptic phenotypes among withheld UCSF cases (Bonferroni-corrected bootstrapped $P$-value $< 0.05$, Supplementary Data 4).

Although the UKBB cryptic phenotype models replicated within UCSF for nearly 40% of the original 50 conditions, their performance (with respect to increased severity among diagnosed cases) was systematically worse (Fig. 2a and b for HHT; Fig. 2d for global comparison; unpaired $T$-test $P$-value $= 0.003$; $N = 13$, which includes all diseases successfully captured in the UCSF dataset with matching UCSF-UKBB cryptic phenotypes and diagnostic codes available in both datasets, see the "Methods" section). The source of this decreased performance is likely multifactorial. For example, the ICD10 encoding used by the UKBB is less granular (see the "Methods" section). This in turn decreases the number of symptoms available for model inference, which can lead to decreased performance. Consistent with this hypothesis, we note that much of the difference in dataset performance disappears when models inferred within the UKBB are applied to the UCSF data (Fig. 2a–c for HHT; see Fig. 2d and

e for a global comparison; unpaired $T$-test $P$-value $= 0.17$; $N = 13$, see above). That said, there are many differences between the clinical datasets in general (sample sizes, population demographics, data provenance, etc.), and it is difficult to disentangle all potential factors. Ideally, cryptic phenotypes would be jointly inferred in the two datasets, allowing their unique information to be shared systematically. However, because our follow up genetic analyses could only be performed in the UKBB (UCSF lacks linked genetic data), all subsequent analyses were performed using models inferred in the UKBB.

Note, the cryptic phenotypes inferred by the UCSF and UKBB models showed variable consistency (as assessed using the coefficient of determination among their predictions within the UCSF dataset, see Fig. 2f and Supplementary Data 4), even for diseases that survived our replication filters. Specifically, 10 of the 18 replicating diseases resulted in phenotype models that generated $r^2$ values among the inferred traits $\geq 0.2$. From this set of 10 conditions, five had a known genetic mechanism that could be directly ascertained within UKBB data (autosomal dominant, X-linked, or phased autosomal recessive); these were selected for follow up rare and common variant genetic analyses (Table 1). Among this group of five, there was still variability in cryptic phenotype consistency (MFS $r^2 = 0.21$ vs. A1ATD $r^2 = 0.89$), which may have affected the performance of downstream analyses.

**Cryptic phenotype validation.** To further validate the inferred cryptic phenotypes, we conducted rare variant association analyses to ensure that these traits could replicate known mechanisms of disease. Because these analyses were conducted for validation rather than discovery, we focused on rare variants that were either: (1) annotated as pathogenic/likely pathogenic (P/LP) in ClinVar[2] or (2) predicted[29] to be loss-of-function (LoF) alleles (referred to as P/LP variants subsequently, see Supplementary Data 6 for full list). Linear regression was performed to assess

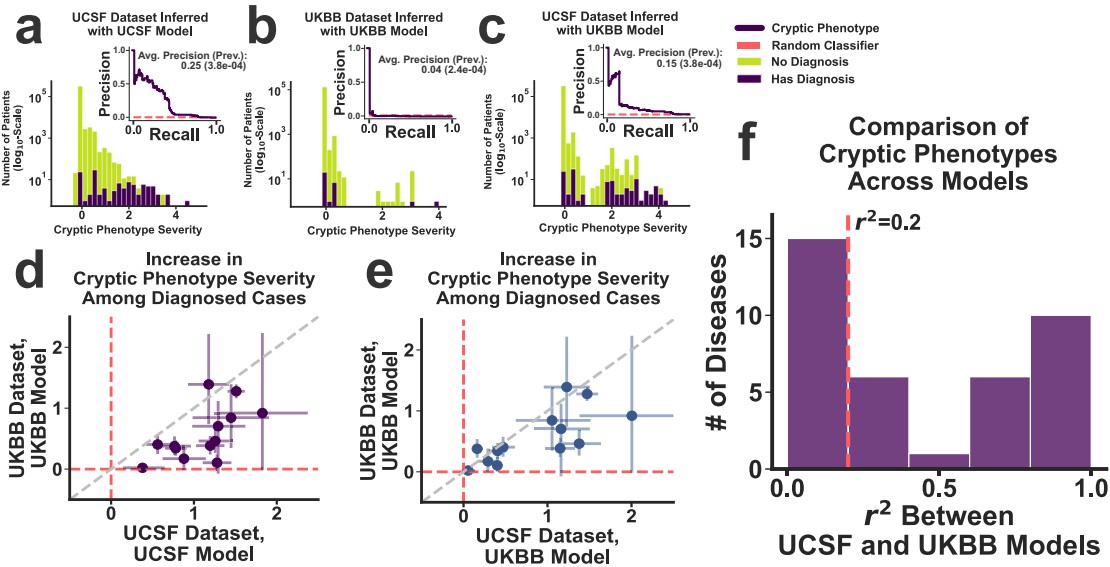

**Fig. 2 Cryptic phenotype inference in the UCSF and UKBB datasets. a** Distribution of HHT cryptic phenotype severity among the subjects in the UCSF testing dataset, stratified by their HHT diagnostic status (green: controls; purple: HHT cases). **a** (inset) Precision-recall curve for the prediction of HHT diagnoses using the cryptic phenotype. The approximate performance of a random classifier is shown in red. Panel **b** displays the same information for the UKBB dataset, which was generated using an independently inferred phenotype model. Panel **c** displays the same information as (**a**), except that the UKBB phenotype model is used to generate the cryptic phenotypes in the UCSF dataset. **d, e** The increase in cryptic phenotype severity among diagnosed cases is displayed jointly for the UCSF and UKBB models/datasets (N = 13 diseases, see main text and Supplementary Data 4). Panel **d** compares the results of the UCSF model (applied to the UCSF dataset; x-axis) with those generated by the UKBB model (applied to the UKBB dataset; y-axis). Panel **e** instead compares the results of the UKBB model after applying it to both the UCSF (x-axis) and UKBB (y-axis) datasets. Error bars in panels **d** and **e** represent 95% confidence intervals for the severity statistics (estimated using bootstrapped re-sampling, $N = 10^5$). Panel **f** Coefficients of determination ($r^2$) among the cryptic phenotypes inferred by the UCSF and UKBB models were estimated using the UCSF dataset. The resulting distribution over this statistic is displayed for the 38 diseases where model fitting was successful in both datasets.

**Table 1 Diseases selected for molecular validation and genomic analysis.**

| Disease name | Abbreviation | Mode of inheritance | Causal genes | Variants analyzed |
|---|---|---|---|---|
| Alpha-1-antitrypsin deficiency | A1ATD | Autosomal recessive | *SERPINA1* | rs28929474 (E342K; Z-allele) |
| Hereditary hemorrhagic telangiectasia | HHT | Autosomal dominant | *ACVRL1; ENG; SMAD4* | P/LP ClinVar variants;novel loss-of-function |
| Marfan syndrome | MFS | Autosomal dominant | *FBN1* | P/LP ClinVar variants;novel loss-of-function |
| Alport syndrome | AS | Autosomal dominant, X-linked | *COL4A3; COL4A4; COL4A5* | P/LP ClinVar Variants;novel loss-of-function |
| Autosomal dominant polycystic kidney Disease | ADPKD | Autosomal dominant | *PKD1; PKD2* | P/LP ClinVar variants;novel loss-of-function |

whether variants were significantly associated with the corresponding cryptic phenotype (see the "Methods" section). For all five disorders in Table 1, carriers of pathogenic, disease-causing variants tended to have more severe cryptic phenotypes (Fig. 3a–c and Supplementary Fig. 6a, b). However, significant variability was observed among P/LP variant carriers, and many of these subjects had few if any apparent symptoms (i.e. cryptic phenotypes = 0, see Fig. 3a–c, insets).

There are multiple factors that may contribute to the phenotypic variability seen among P/LP carriers. First, EMR data is an imperfect proxy for an individual's true symptoms, and it is certainly possible that missing information accounts for a significant fraction of this variability. Second, some of the P/LP variants may be misclassified. Consistent with this hypothesis, we note that variants that were flagged due to annotation issues (see the "Methods" section) tended to have smaller effect sizes (see Fig. 3a–c). Third, confirmation bias could result in the documentation of symptoms that would otherwise be left out of the EMR (ex: epistaxis in a known case of HHT), resulting in inflated cryptic phenotypes among diagnosed cases (see Fig. 3a–c, inset and Fig. 3d, top).

Alternatively, inflated cryptic phenotypes would also be observed if only the most severely affected individuals receive a rare disease diagnosis. In other words, the inflation of cryptic

phenotypes seen among diagnosed pathogenic variant carriers could instead be driven by ascertainment bias at the level of disease diagnosis (Fig. 3d, bottom). The identification of specific genetic modifiers could help differentiate between these two models (Fig. 3e). Since it is impossible for a disease diagnosis to alter an individual's genotype, an association between common variation that modifies disease expressivity and the diagnosis itself is only consistent with a model in which symptom severity affects disease ascertainment (Fig. 3e, bottom). Therefore, investigating a role for common variation in cryptic phenotype severity can serve two purposes. It can identify background genetic variation that may modify symptom severity, and it can also help distinguish different types of bias that may be present within EMR data.

**Common variation is associated with cryptic phenotype variability.** To identify potential common variant modifiers of cryptic phenotype severity, we first divided the UKBB into two subsets for each disease–trait pair, which we refer to as the training and target cohorts. The training cohorts included unrelated subjects of similar genetic ancestry (Caucasian). P/LP carriers, diagnosed rare disease cases, and all their 3rd degree or closer relatives were specifically excluded from these subsets (see the "Methods" section). The training cohorts were used for

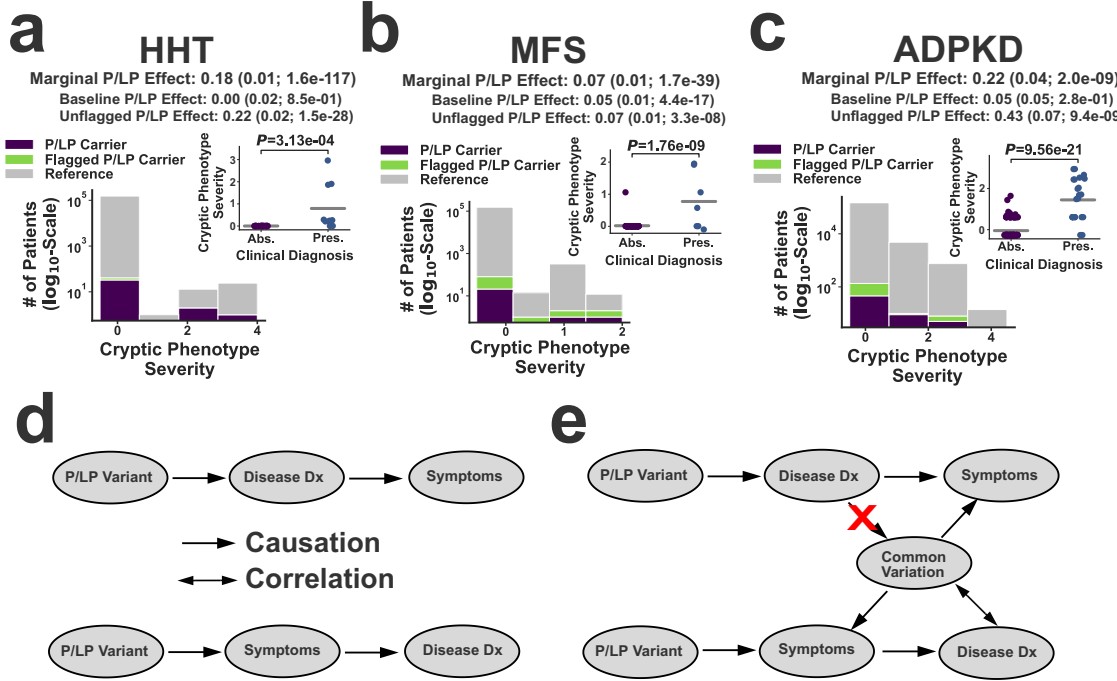

**Fig. 3 Exome-sequencing validation of the inferred cryptic phenotypes.** Panels **a–c** Histograms depicting the distribution of cryptic phenotype severity across three different genotypes in the UKBB: reference (gray), P/LP Carrier (purple), and Flagged P/LP Carrier (green). The marginal P/LP cryptic phenotype effect sizes (includes flagged and unflagged variants) are provided at the top of each panel. The decomposed baseline and unflagged variant effects are displayed below the marginal effects. Parentheses contain the standard errors and P-values for the effects. These statistics were estimated using linear regression and two-sided T-tests (see the "Methods" section). The insets display the cryptic phenotype severity estimates among the P/LP carriers for each condition, stratified by whether the rare disease diagnosis is absent (Abs.) or present (Pres.). Gray bars represent the mean values within each diagnostic class, and P-values were computed using linear regression (with Clinical Diagnosis included as a predictor) and two-sided T-tests. No adjustments were made for multiple testing. **a** Hereditary hemorrhagic telangiectasia (HHT; $N = 153,182$ independent subjects). **b** Marfan syndrome (MFS; $N = 153,182$ independent subjects). **c** Autosomal dominant polycystic kidney disease (ADPKD; $N = 153,182$ independent subjects). **d** Illustration of the two biases that could lead to increased cryptic phenotypes among diagnosed carriers. Top: post-diagnosis confirmation bias. Bottom: pre-diagnosis ascertainment bias. **e** Common variant modifiers could be used to distinguish between these competing models, as common variation would only be correlated with disease diagnosis under the ascertainment bias scenario.

genome-wide association analyses and polygenic prediction model inference ($N = 294,133$–$308,381$). Alternatively, the target cohorts contained subsets of unaffected and unrelated subjects of similar ancestry (to provide power for replication; $N = 32,682$–$34,265$) plus all the individuals affected by the monogenic disease of interest (after removing any 3rd degree or closer relatives among this subset; $N = 166$–$17,163$).

The results of the genome-wide association analyses conducted on the training cohorts are summarized using Quantile–Quantile plots in Fig. 4. For three of the five disorders (A1ATD, AS, and ADPKD, Fig. 4a, d, and e), the common variant heritability was significantly increased from zero, consistent with a role for genetic background effects in phenotypic variability. For two disorders (HHT and MFS), the heritability was indistinguishable from zero, even though there was evidence for test-statistic inflation at low minor allele frequencies. The etiology of this test-statistic inflation is unclear. It may be driven by the non-Gaussian nature of the cryptic phenotype distribution. Alternatively, residual population structure exacerbated by large sample sizes[30,31] could also result in test statistic inflation (see Supplementary Table 2 for genomic inflation factors re-scaled for a smaller sample size). Regardless, these results do not exclude a role for common variants in the phenotypic heterogeneity of these traits. The cryptic phenotype models for both HHT and MFS showed reduced consistency across datasets ($r^2 = 0.23$ and $r^2 = 0.21$ for HHT and MFS), suggesting that improved modeling may be able to infer cryptic phenotypes with better performance.

Ultimately, polygenic prediction models were inferred (using individual level data[32], see the "Methods" section) for the cryptic phenotypes with non-zero heritability, specifically those belonging to A1ATD, AS, and ADPKD (models provided in Supplementary Data 7–9). These models were then used to impute polygenic scores (PGS) into the target cohorts so that the detected genetic effects could be replicated and validated.

**CPA identifies common variation associated with A1ATD severity.** Alpha-1-antitrypsin deficiency (A1ATD) is a relatively common genetic disorder that leads to early-onset emphysema, liver disease, and auto-inflammatory symptoms[33]. The Pi*Z allele (*rs28929474*) in *SERPINA1* is the most common cause of severe A1ATD, although the penetrance of this variant is incomplete. The clinical manifestations associated with the Pi*Z allele are known to depend heavily on genotype (the A1ATD phenotype is much more severe among Pi*ZZ homozygotes vs. Pi*MZ heterozygotes) and environmental background effects (smoking, alcohol use, etc.)[34]. Common variant modifiers likely also play a significant role[35]. Using the cryptic phenotype approach, we aimed to further investigate the potential effects of background genetic variation on A1ATD severity.

The GWAS conducted on the A1ATD cryptic phenotype (Fig. 5a, Manhattan plot) detected three genome-wide significant loci. Not surprisingly, they have all been previously linked to chronic pulmonary disease, lung function, and smoking[36] (Supplementary Table 3). These results are consistent with the

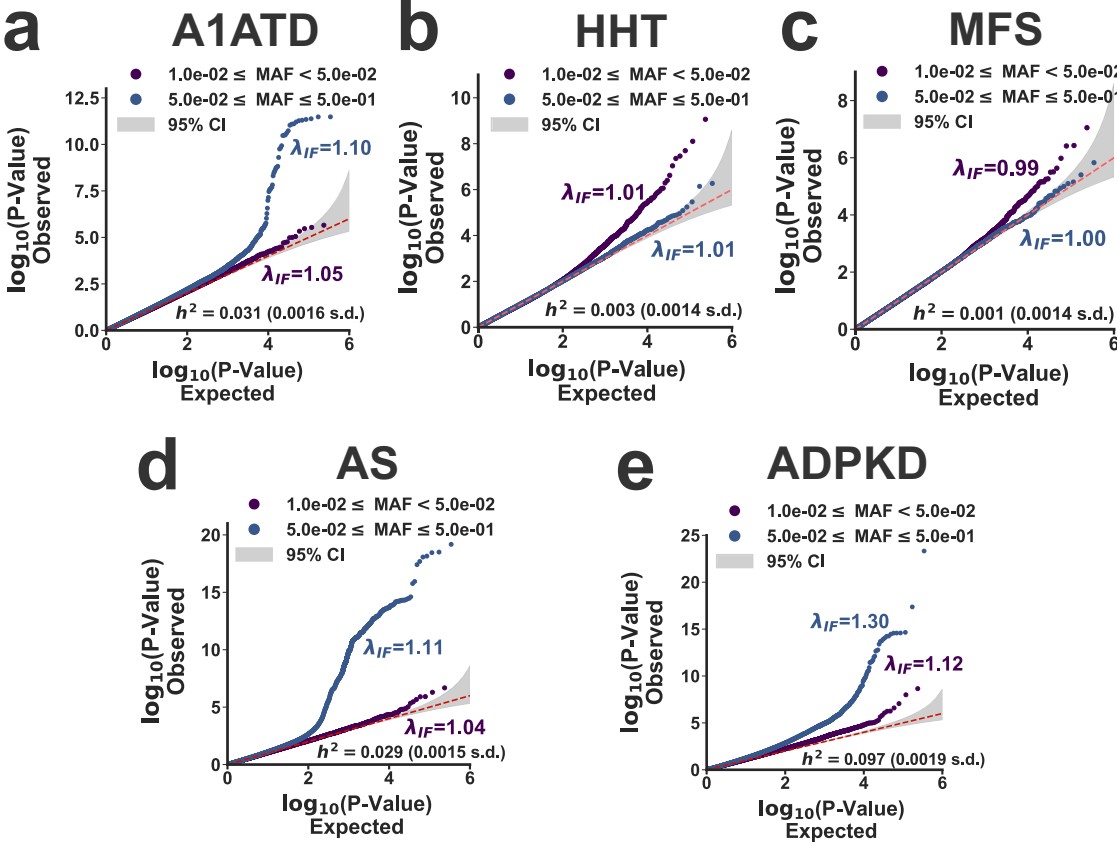

**Fig. 4 Common variation associated with cryptic phenotype severity. a–e** Each panel displays the observed versus expected *P*-value quantiles for the cryptic phenotype genome-wide association statistics, stratified by allele frequency (purple: 0.01≤AF < 0.05; blue 0.05 ≤ AF ≤ 0.50). Genomic inflation factors ($\lambda_{IF}$) are provided in addition to the common variant heritability estimates ($h^2 \pm$ std errors; estimated using the LDAK toolkit[76], see the "Methods" section). The dashed red lines indicate the expected quantile–quantile relationship under the null model of no association between the variants and the phenotype. The shaded gray areas represent the 95% confidence interval for this expected relationship. **a** α-1-Antitrypsin deficiency (A1ATD; *N* = 294,133 independent subjects). **b** Hereditary hemorrhagic telangiectasia (HHT; *N* = 308,381 independent subjects). **c** Marfan syndrome (MFS; *N* = 308,350 independent subjects). **d** Alport Syndrome (MFS; *N* = 308,088 independent subjects). **e** Autosomal-dominant polycystic kidney disease (ADPKD; *N* = 308,095 independent subjects).

strong effects that smoking is known to have on A1ATD severity[34]. To further investigate, we examined the interaction between smoking history (measured as reported pack-years; UKBB Data Field: 20161) and the Pi*Z allele using the inferred cryptic phenotype. Symptom severity was substantially elevated among heavy smokers, both within and across the pathogenic genotypes (Fig. 5b and b inset for Pi*MZ and Pi*ZZ genotypes, respectively). The cryptic phenotype polygenic score (PGS) was strongly associated with smoking history (Fig. 5c), and after regressing out the effects of smoking, the PGS remained associated with phenotypic severity ($\beta_{PGS} = 0.02$; *P*-value = $1.6 \times 10^{-11}$). This suggests that the PGS may capture background effects that are independent of smoking history. However, it is important to note that the relationship among smoking history, *SERPINA1* genotype, and polygenic load is likely complex. For example, Fig. 5e depicts the PGS effects on cryptic phenotype severity among pathogenic variant carriers, stratified by smoking history and genotype. Notably, the PGS effect varies considerably depending on whether an individual has ever smoked, particularly among Pi*ZZ carriers ($\beta_{PGSxPi*ZZ} = 0.41$ among smokers vs. $\beta_{PGSxPi*ZZ} = -0.13$ among non-smokers; LR test for smoking-by-PGS interaction effects: *P*-value = $2.1 \times 10^{-9}$). The source of this variability is uncertain, but we hypothesize that it may be partially driven by smoking cessation/

abstinence among more severely affected pathogenic variant carriers (Supplementary Fig. 7b).

To further validate the inferred PGS, we tested whether polygenic load was significantly associated with A1ATD diagnoses. Unfortunately, structured diagnostic data for A1ATD is not available in the UKBB medical records, but A1ATD diagnoses (as provided by a physician) were ascertained as part of a survey that was conducted among the study participants (UKBB Data Field: 22152). Consistent with ascertainment bias at the level of disease diagnosis (see Fig. 3d and e), the cryptic phenotype PGS was significantly associated with the risk for A1ATD diagnosis (Firth-corrected logistic regression LR test; $\beta_{PGS} = 0.50$; *P*-value = 0.01), which was compounded by the large (and expected) effects of the pathogenic genotypes themselves ($\beta_{PiZZ} = 8.98$; *P*-value = $1.5 \times 10^{-26}$; $\beta_{PiMZ} = 4.75$; *P*-value = $2.3 \times 10^{-15}$).

To determine if increased polygenic load translated to other outcomes, we examined the variability in age-of-onset for chronic obstructive pulmonary disease (COPD; Data Field: 42016) among the different genotypes within our target cohort. Consistent with prior knowledge, both the Pi*MZ and Pi*ZZ genotypes resulted in more frequent and earlier onset COPD ($\beta_{PiZZ} = 2.8 \pm 0.2$, *P*-value = $2.2 \times 10^{-25}$; $\beta_{PiMZ} = 0.16 \pm 0.06$, *P*-value = 0.02; Cox-Proportional Hazards regression, see the "Methods" section). Furthermore, smoking history (in pack-years) had a profound

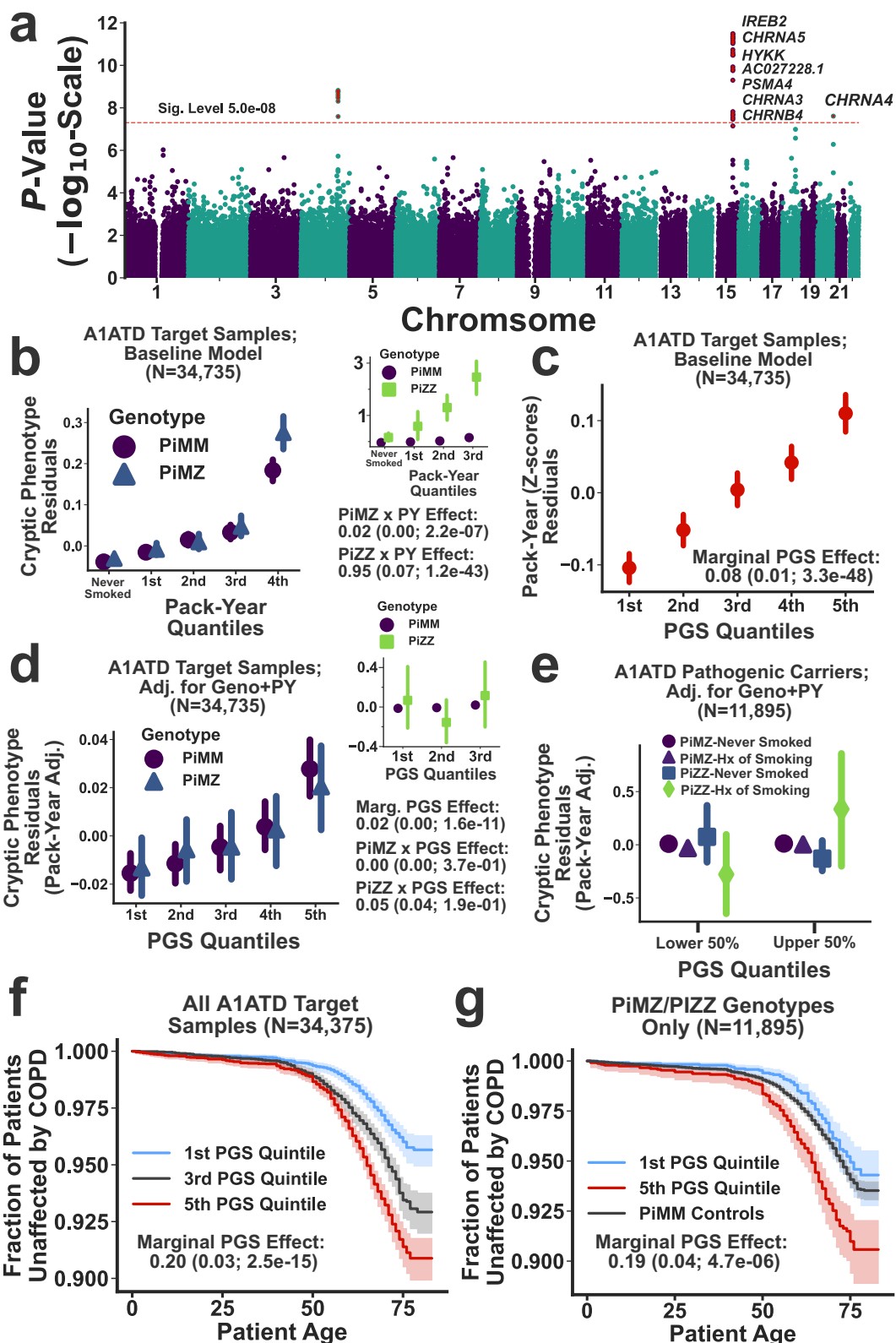

effect on COPD onset ($\beta_{Smoke} = 0.47 \pm 0.01$; $P$-value $= 6.8 \times 10^{-239}$), which included significant smoking-by-genotype interaction effects (LR test $P$-value $= 1.9 \times 10^{-5}$). After correcting for smoking history, the cryptic phenotype PGS had a significant, additive effect ($\beta = 0.20 \pm 0.03$; $P$-value $= 2.5 \times 10^{-15}$, Fig. 5e), which persisted even when limiting the analyses to only those

individuals that carry the Pi*MZ/Pi*ZZ genotypes ($\beta_{PGS} = 0.19 \pm 0.04$; $P$-value $= 4.7 \times 10^{-6}$; see Fig. 5f). This additive PGS effect also replicated in spirometry measurements (Supplementary Fig. 7c). Note, we performed this analysis using only those subjects with the Pi*ZZ genotype, but the sample size ($N = 102$) was likely too small to detect a significant effect ($\beta_{PGS} = 0.17 \pm$

**Fig. 5 Cryptic phenotype-associated genetic variation modifies A1ATD severity. a** Manhattan plot displaying the genome-wide association statistics as a function of chromosomal position. Genes were assigned to loci using FUMA[75]. The $5 \times 10^{-8}$ significance threshold is displayed as a dashed red line, and significant loci are highlighted with red stars. **b** Cryptic phenotype (CP) residuals are stratified by the Pi*MZ/Pi*MM genotypes and plotted as function of pack-year quantiles. Inset: CP residuals plotted against pack-year quantiles, now stratified by the Pi*ZZ/Pi*MM genotypes. In both panels, the points represent the mean value within each quantile, and the error bars represent the 95% confidence intervals (CIs) for the mean (obtained through bootstrapped re-sampling, $N = 1000$). The association statistics for the genotype x smoking interaction terms are included below the inset (estimated using linear regression and two-sided $T$-tests). **c** Smoking history (expressed as pack-years) is plotted against PGS quantiles (points/error bars indicate quantile means/95% CIs). **d** CP residuals, after adjusting for baseline covariates, genotype, and smoking history, are plotted against PGS quantiles and stratified by the Pi*MZ/Pi*MM genotypes. The inset displays the same information but now stratified by the Pi*ZZ/Pi*MM genotypes. Both panels depict the quantile means and their associated 95% CIs. The association statistics for the PGS effects (estimated using linear regression and two-sided $T$-tests) are included below the inset. **e** CP residuals within the upper and lower 50th percentiles of the PGS distribution are stratified by both genotype and smoking history (points/error bars represent subset means/95% CIs). **f** Kaplan–Meier curves for COPD onset after stratifying the target cohort according PGS quintiles. **g** Same as in **f**, except only subjects with the Pi*MZ/Pi*ZZ genotypes are included. The PGS effect size and association statistics (computed using a Cox Proportional-Hazards model, see the "Methods" section) are provided for the subjects depicted in **f** and **g**. The shaded regions represent the 95% CIs for the survival curves. The summary statistics reported in this figure were not adjusted for multiple testing.

0.24; $P$-value $= 0.46$, see Supplementary Fig. 7d). In total, these results indicate that the cryptic phenotype for A1ATD replicates much of the known architecture for A1ATD[34] while also identifying common genetic variation that modifies symptom expression and severity.

**CPA identifies putative modifiers of monogenic kidney disease.** Alport syndrome and autosomal-dominant polycystic kidney disease represent two of the most common forms of hereditary kidney disease[37,38], although their underlying molecular pathophysiology is distinct. Alport syndrome is a genetically heterogenous Type IV collagenopathy linked to the *COL4A3*, *COL4A4*, and *COL4A5* genes. The collagen isoforms produced by these genes play an integral role in maintaining basement membrane integrity within the glomerulus[39], cochlea[40], and eye[41]. In its mildest form (often referred thin basement membrane nephropathy[42]), the disorder is associated with persistent hematuria that uncommonly progresses to chronic kidney disease. In these cases, the disease is typically caused by heterozygous pathogenic variants located within any of the three causative genes. In the severe form, the disease is characterized by end-stage renal disease, hearing loss, and vision abnormalities. Such individuals typically harbor hemizygous variants in *COL4A5* (X-linked) or biallelic pathogenic variants in *COL4A3/COL4A4*[37]. Alternatively, ADPKD is linked to the *PKD1* and *PKD2* genes, which encode two integral membrane proteins that play complex roles in $Ca^{2+}$ regulation and ciliary functioning[38]. Phenotypically, ADPKD leads to chronic kidney disease more consistently, although there is again a great deal of variability in age of onset and rate of progression[43]. Moreover, extra-renal manifestations are present in a significant fraction of ADPKD patients, and such symptoms include other organ cysts, vascular aneurysms, hernias, and bronchiectasis[44].

To investigate a role for common genetic variation in AS and ADPKD variability, we conducted GWAS on their respective cryptic phenotypes. The results are displayed in Fig. 6a and d. For AS, three loci reached genome-wide significance (see Supplementary Table 4). Interestingly, the locus on chromosome 19 has previously been linked to hematuria[45], and the locus on chromosome 13 is located within the intron of another Type IV collagen isoform (*COL4A2*). This locus has also been linked to neurovascular phenotypes[36]. The third locus is proximal to the MHC region on chromosome 6, and due to complex linkage disequilibrium, it has been associated with many disparate phenotypes[36]. The GWAS for ADPKD uncovered 30 independently associated loci (see Fig. 6d and Supplementary Table 5), most of which have been previously linked to kidney disease and blood pressure regulation[36].

After performing the genome-wide association analyses, prediction models were constructed to capture the global effects of polygenic load on cryptic phenotype severity. With respect to AS, the inferred PGS had a significant marginal effect in the withheld target cohort ($\beta_{PGS} = 0.03 \pm 0.00$; $P$-value $= 9.8 \times 10^{-14}$), which was more pronounced among the P/LP carriers ($\beta_{PGSxP/LP} = 0.09 \pm 0.03$; $P$-value $= 0.002$). Diagnostic data for AS is not available within the UKBB, so we instead focused on two critical outcomes related to the disease: recurrent and persistent hematuria (UKBB Data Field: 132002) and end-stage renal disease (ESRD; UKBB Data Field: 42026). P/LP variants in AS genes were significantly associated with both outcomes (persistent hematuria: $\beta_{P/LP} = 1.06$, $P$-value $= 0.04$; ESRD: $\beta_{P/LP} = 1.70$, $P$-value $= 3.7 \times 10^{-4}$; Firth-corrected logistic regression), although these effects were less apparent within the age-of-onset data (see Supplementary Fig. 8d, e). The cryptic phenotype PGS was marginally associated with persistent hematuria ($\beta_{PGS} = 0.31$; $P$-value $= 0.007$; Firth-corrected logistic regression), an effect that was also apparent when modeling the age-of-onset ($\beta_{PGS} = 0.31 \pm 0.12$; $P$-value $= 0.03$; Cox proportional hazards model, see Fig. 6c). Unfortunately, there were too few persistent hematuria cases to determine if there was a significant interaction effect between the polygenic background and P/LP variants ($\beta_{PGSxP/LP} = -0.00 \pm 0.62$; $P$-value $= 0.86$). Note, there was no evidence that the cryptic phenotype PGS for AS was associated with ESRD ($\beta_{PGS} = 0.03$; $P$-value $= 0.82$; Firth-corrected logistic regression). However, it was significantly predictive of urine microalbuminuria ($\beta_{PGS} = 3.10 \pm 1.36$; $P$-value $= 0.023$; Supplementary Fig. 8c), suggesting that the PGS correlates with glomerular dysfunction.

As was the case for AS, the PGS constructed using the cryptic phenotype for ADPKD was again strongly associated with the trait in the target cohort ($\beta_{PGS} = 0.06 \pm 0.00$; $P$-value $= 3.5 \times 10^{-134}$), and like before, the effect was more pronounced among the P/LP carriers ($\beta_{PGSxP/LP} = 0.15 \pm 0.05$; $P$-value $= 0.003$; see Fig. 6e). In contrast to AS, diagnostic data for ADPKD is available within the UKBB. As expected, P/LP carrier status was strongly associated with Mendelian disease diagnoses (Firth-corrected logistic regression; $\beta_{P/LP} = 4.47$; $P$-value $= 3.5 \times 10^{-31}$), but the inferred PGS had no discernable marginal ($\beta_{PGS} = 0.02$; $P$-value $= 0.67$) or interaction ($\beta_{PGSxP/LP} = 0.37$; $P$-value $= 0.13$) effects. A substantial fraction of P/LP carriers in the UKBB were diagnosed with ADPKD (specifically, 35% with polycystic kidney disease and 48% with cystic kidney disease in general), so it is possible that these diagnoses lacked the variability needed to detect interaction effects. Therefore, we also examined if the cryptic phenotype PGS affected ADPKD onset and rate-of-progression.

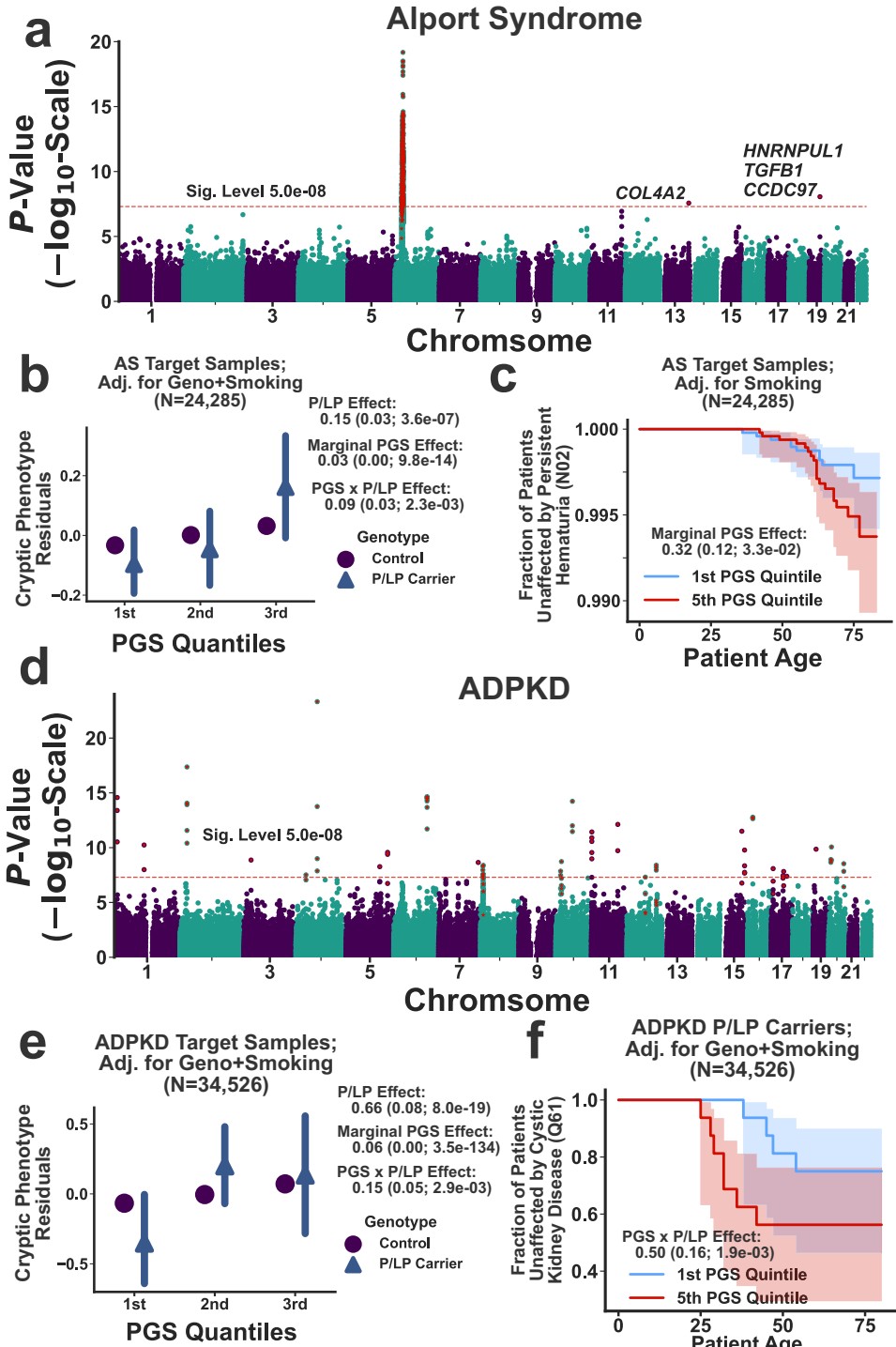

Cystic kidney disease onset (UKBB Data Field: 132533) was modeled as a function of both P/LP carrier status and polygenic load. As expected, ADPKD P/LP carriers were at high risk for early-onset cystic kidney disease (Cox proportional-hazards regression model; $\beta_{P/LP} = 3.73 \pm 0.29$; $P$-value $= 3.2 \times 10^{-36}$), consistent with the known pathophysiology of the disorder. Interestingly, there was a significant interaction effect between the cryptic phenotype PGS and P/LP carrier status ($\beta_{PGSxP/LP} = 0.50 \pm 0.16$; $P$-value $= 0.002$; see Fig. 6f), consistent with a model in which polygenic load modulates ADPKD severity. Because ESRD is the downstream effect of severe cystic kidney disease, we used

the onset of this phenotype as a proxy for ADPKD progression. Once again, P/LP carrier status had a profound effect on ESRD onset ($\beta_{P/LP} = 3.91 \pm 0.37$; $P$-value $= 7.0 \times 10^{-19}$, see Supplementary Fig. 9d), and there was again a significant interaction effect between P/LP carrier status and polygenic load ($\beta_{PGSxP/LP} = 0.50 \pm 0.21$; $P$-value $= 0.02$; see Supplementary Fig. 9e). To further verify this effect, we estimated[46] the glomerular filtration rate (eGFR) within our target cohort. P/LP carriers had significantly lower eGFR values ($\beta_{P/LP} = -15.2 \pm 2.0$; $P$-value $= 7.2 \times 10^{-15}$), and there was again a significant interaction effect between carrier status and the inferred PGS ($\beta_{PGSxP/LP} = -3.8 \pm 1.3$; $P$-value $=$

**Fig. 6 Common variant modifiers of monogenic kidney disease revealed through cryptic phenotype analysis. a** Manhattan plot displaying the genome-wide association statistics for the AS cryptic phenotype as a function of chromosomal position. Genes were assigned to loci using FUMA[75]. The $5 \times 10^{-8}$ significance threshold is displayed as a dashed red line, and significant loci are highlighted with red stars. **b** AS cryptic phenotype (CP) residuals, after adjusting for baseline covariates, P/LP genotype, and pack-years, are plotted against PGS quantiles and stratified by the P/LP carrier status. The points represent the mean value within each quantile, and the error bars represent the 95% confidence intervals (CIs) for the mean (obtained through bootstrapped re-sampling, $N = 1000$). The association statistics for the P/LP variants, the PGS, and their interaction effects are included to the right (estimated using linear regression and two-sided $T$-tests). **c** Kaplan–Meier curve for Persistent Hematuria (see the "Methods" section) is stratified by PGS quintile. The shaded regions represent the 95% CIs for the survival curves. The marginal PGS effect and summary statistics were estimated using Cox proportional-hazards regression (see the "Methods" section). **d** Manhattan plot displaying the genome-wide association statistics for the ADPKD cryptic phenotype as a function of chromosomal position. **e** ADPKD cryptic phenotype (CP) residuals, after adjusting for baseline covariates, P/LP genotype, and smoking status (see the "Methods" section), are plotted against PGS quantiles and stratified by the P/LP carrier status. The points and error bars represent the mean value within each quantile and the 95% CIs, respectively. The association statistics for the P/LP variants, the PGS, and their interaction effects are included to the right (estimated using linear regression and two-sided $T$-tests). **f** Kaplan–Meier curve for cystic kidney disease onset stratified by PGS quintile. Note, only the P/LP carriers are depicted; however, the PGSxP/LP interaction effects were computed using the complete target cohort (effect size and summary statistics estimated using Cox proportional-hazards regression). The summary statistics reported in this figure were not adjusted for multiple testing.

---

0.005; see Supplementary Fig. 9c). Overall, these results suggest that polygenic burden is associated with worse outcomes among P/LP carriers, consistent with a role for common variant effects in modifying ADPKD disease severity.

## Discussion

Cryptic phenotype analysis (CPA) uses statistical modeling to infer quantitative traits that summarize disease-related phenotypic variability. These traits are estimated using symptoms documented in the electronic medical record (EMR). In the current study, we used cryptic phenotypes to identify common genetic variants putatively associated with Mendelian disease severity. Our approach relies on two assumptions. First, the disease of interest must represent the severely affected extreme of a spectrum of phenotypic variation. Second, shared genetic factors must drive this variability within the mildly and severely affected individuals. If true, then modifier effects should be detectable within a subset of the population that extends beyond the rare disease cases. Consistent with this hypothesis, we used CPA to identify common genetic variation associated with Mendelian disease severity, even though our initial association analyses were conducted in populations that were specifically depleted for the rare disease cases. The results suggest that shared genetic factors drive variability across the full range of cryptic phenotype severity and predict a specific role for common variation in the genetic architecture of Mendelian disease-associated traits. Nevertheless, there are multiple avenues for further investigation.

CPA has several attractive properties. First, it performs latent phenotype inference using an unsupervised generative model (see the "Methods" section), thereby directly estimating quantitative traits that summarize symptom variability. Second, its model-based nature allows cryptic phenotypes to be directly imputed into new datasets, albeit only if the observed data is encoded in the same format. Third, the probability model underlying CPA is modular (i.e. composed of conditionally independent components), so it can be easily extended to incorporate new data types and assumptions. For example, our study used structured EMR data (ICD10 diagnostic codes) to capture disease-related phenotypic variability. However, these data likely provide an incomplete view of clinical heterogeneity. Because CPA is performed using a fully generative model, additional datatypes such as laboratory results or even unstructured clinical data could be incorporated into the framework by specifying new data-generating functions. The inclusion of such datatypes could in turn result in more accurate latent phenotypes, ultimately increasing power for downstream analyses. That said, simpler heuristic[9] and discriminative[47] approaches also exist for quantifying disease-

related variability, and additional work is needed to determine if and/or when such indirect approaches can be used to perform the types of analyses described in this study.

The results from the common variant association analyses demonstrate that polygenic load likely plays a role in Mendelian disease variability. These polygenic effects were detected at the level of the cryptic phenotypes themselves (Figs. 5d, e, 6b, and e), but they were also apparent when examining outcomes known to be associated Mendelian disease severity (spirometry measurements, glomerular filtration rate, symptom age-of-onset, etc.). Although the results replicated across Mendelian diseases (i.e. polygenic load was consistently associated with more severe outcomes), it is difficult to replicate the results across datasets, as a unique combination of information (structured EMR data, genome-wide common variation, and exome sequencing data) is required. Biobanks with linked medical and genetic data are becoming increasingly common[48–52], so the types analyses described here will soon become easier to perform and replicate. In addition, these new biobanks will contain a wealth of genetic and phenotypic diversity. Although this information will certainly improve our understanding of human genetics, it will come at the cost of increased phenotypic complexity, particularly as more types of clinical information (laboratory results, unstructured notes, imaging, etc.) become available. We anticipate that methods like CPA, which summarize and simplify high-dimensional phenotypes, will play an important role in analyzing these expanding datasets.

The current study uses CPA to identify common genetic variation associated with Mendelian disease severity and outcomes. The general approach, however, could be applied to diseases with even more complex genetic architectures, provided that they are associated with a diverse array of clinical findings (e.g. systemic lupus erythematosus). In addition, the analyses presented here focus on distilling disease heterogeneity into a single quantitative trait. However, it is also possible to use a method like CPA to decompose disease-related variability into multiple sub-phenotypes, an approach that is already being leveraged to investigate the genetic architecture of conditions like Type 2 diabetes[53], psychiatric illness[54], and asthma[55]. We also note cryptic quantitative traits likely have applications outside of genetic association analyses. For example, they could be used to assist with rare variant annotation (see Fig. 3a–c for examples) or identify environmental modifiers (Fig. 5b).

In summary, cryptic phenotype analysis systematically estimates quantitative traits that capture spectrums of phenotypic variation using qualitative symptom data. By applying this approach to Mendelian diseases, we were able to identify putative

modifiers of disease severity. The work described here builds upon a growing number of studies[9,56,57] that demonstrate the utility of applying statistical models of human phenotypes to population-scale medical record and genetic datasets. However, the wider adoption of CPA and similar methods will require the development of flexible and robust statistical models that can reliably summarize high-dimensional phenotypes isolated from increasingly complex clinical datasets. In addition, more work is needed to explore the utility and limitations of these approaches, particularly across diverse diseases and populations. Nevertheless, as biobank-scale datasets become more common, we anticipate that CPA and similar methods will continue to shed light onto the genetic complexity of high-dimensional human phenotypes.

## Methods

**Clinical datasets.** Phenotypic analyses were conducted using the University of California San Francisco De-Identified Clinical Data Warehouse (UCSF-CDW)[58], a database of structured health information that is made available to UCSF researchers free-of-charge. The data was captured for use on 31 May 2019 and includes roughly 8 years of clinic visits and inpatient hospitalizations (see Supplementary Methods). Following capture, patient demographic data was aligned to the International Classification of Disease, Tenth Revision, Clinical Modification (ICD10-CM) diagnostic codes available within the medical encounters. The individual diagnostic codes were simplified by collapsing multiple appearances of each code into a single value (at-least-one binarization), enabling the full set of diagnostic codes specific to each patient to be stored as a sparse, binary array. The ICD10-CM codes were filtered according to multiple criteria, which are described in the Supplementary Methods. This generated a dataset containing 10,483 ICD10-CM codes aligned to 1,204,212 patients. This is subsequently referred to as the UCSF-ICD10-CM dataset.

The UCSF-ICD10-CM was further processed in two ways. First, the ICD10-CM codes were transformed into human phenotype ontology (HPO)[26] terms using a customized mapping, the construction of which is outlined below and in the Supplementary Methods (resulting dataset denoted UCSF-HPO). This alignment resulted in a global diagnostic matrix encoding 1674 HPO symptoms. Second, we translated the ICD10-CM codes into the ICD10 terminology utilized by the UK Biobank (ICD10-UKBB), taking advantage of the fact that the UKBB encoding is a less granular subset of the ICD10-CM (details regarding the precise translation can be found within our vLPI software package available on Github[59]). This processed dataset is subsequently referred to as UCSF-ICD10-UKBB. The UCSF-ICD10-UKBB dataset was also translated into HPO terms (denoted UCSF-HPO-UKBB). These less granular datasets contained 4933 and 1423 diagnostic terms respectively.

The UK Biobank (UKBB) is a collection of ≈500,000 middle-aged British adults who have received extensive genotyping and phenotyping[23]. The bulk UKBB dataset was downloaded on 22 January 2020 using the software provided by the organization. Following download, the raw data file was parsed, isolating demographic variables of interest and collapsing main/secondary inpatient summary diagnoses into a single data value (using at-least-one binarization). The resulting diagnostic codes were filtered according to multiple criteria (see Supplementary Methods), resulting in a 1:1 correspondence between the diagnostic codes available within the UKBB and the UCSF-ICD10-UKBB datasets. These ICD10 codes were then translated into HPO terms. The full UKBB dataset (after removing withdrawn subjects; $N = 502,488$) was used for cryptic phenotype inference, but the subjects were also filtered according to recommended best practices for genetic analyses[23,60]. Filtering resulted in the following two subsets: (1) 485,014 subjects (with exome data, $N = 199,234$) that remained after removing individuals whose genetic data is likely to be confounded by artifact (UKBB-Full), and 2) 342,796 unrelated subjects (with exome data, $N = 153,182$) of likely Western European (Caucasian) ancestry (UKBB-Unrelated). Further details regarding this processing can be found in the Supplementary Methods.

Because the UCSF-CDW and UKBB were both used for phenotype model inference and evaluation, the datasets were a priori divided each into training and testing subsets. To ensure that the testing datasets contained positive cases for each rare disease included in our analysis, distinct training and testing subsets were generated for every disorder. The subsets were constructed by randomly subsampling 75% of the data for training and 25% for testing while maintaining an equal ratio of diagnosed rare disease cases in each. All model inference and preliminary analyses were performed using the training datasets, while the testing datasets were only used for the final evaluation of cryptic phenotypes (see below, Fig. 2d and e, and Supplementary Methods).

**Aligning rare diseases to structured medical data.** Based on previous work[9,61–63], we integrated multiple biomedical ontologies and terminologies to map rare diseases and their symptoms to structured medical data (i.e. diagnostic billing codes). To generate a set of rare diseases for analysis, we first used the Human Disease Ontology[64] to obtain mappings between the Online Mendelian Inheritance in Man (OMIM) database[65] and the ICD10-CM terminologies.

Building on previous work[62], we curated the OMIM-to-ICD10-CM alignments, selecting and grouping ICD10-CM codes that reliably mapped to a single or homogenous set of OMIM diseases, ensuring that the disorders were also annotated within the Human Phenotype Ontology[26]. This resulted in 166 rare, Mendelian conditions that were aligned to both the HPO and ICD10 terminologies (Supplementary Fig. 1). The 166 diseases were sorted according to their diagnostic prevalence in the UCSF-CDW; 50 disorders were selected for follow up testing (see Supplementary Methods; listed in Supplementary Data 1).

The HPO symptoms themselves were aligned to the ICD10-CM terminology in an automated fashion by integrating the information contained within multiple biomedical ontologies[66–69]. Details regarding the alignment are provided in the Supplementary Methods. This resulted in 1674 unique alignments between HPO terms and ICD10-CM codes (Supplementary Data 2). We assessed their performance by using them as features in a rare disease diagnosis prediction task (Supplementary Fig. 2). We found that prediction models constructed from the annotated[26], ICD10-CM-aligned HPO terms had performances that were similar to models constructed using the complete ICD10-CM codebook (see Supplementary Table 1).

**Cryptic phenotype analysis.** Cryptic phenotype analysis (CPA) refers to the process by which a set of symptoms is used to infer a univariate, quantitative trait that captures the clinical heterogeneity observed within a disease of interest. This quantitative but cryptic phenotype can be used to assess clinical variability in both the diagnosed cases and the more general population, enabling the types of analyses described above. CPA consists of two stages. In the first, the symptoms annotated to a particular disease are decomposed into a low-dimensional set of quantitative, latent phenotypes. In the second stage, the trait that best captures disease morbidity (i.e. its symptom expressivity) is identified, since multiple latent traits are often recovered from a single symptom matrix. Below, we briefly outline the two stages of CPA. A more detailed description is provided in the Supplementary Methods.

*Latent phenotype inference.* Consider the set of $K$ symptoms that are associated with some rare disease of interest, and furthermore, assume that these symptoms are binary (present/absent) and permanent (i.e. once diagnosed, they do not resolve). Let $S_{i,j}$ denote the status of the $j$th symptom in the $i$th subject such that $S_{i,j} = 1$ indicates that the patient has been diagnosed with this symptom. Furthermore, let $S$ denote an $N \times K$-dimensional matrix of symptom diagnoses such that the $i$th row of the matrix (denoted $S_i$) contains the diagnoses for subject $i$. Finally, let $Z$ denote an $N \times L$-dimensional matrix of latent phenotypes, where each column represents the magnitude (i.e. severity) of an independent latent phenotype. We modeled the joint likelihood of the disease symptoms and latent phenotypes according to

$$P(S, Z|\theta) = f(Z;\theta) \times P(Z) \qquad (1)$$

where $f(Z; \theta)$ is the symptom risk function (defined by the parameter set $\theta$) that maps the latent phenotypes onto the matrix of symptom probabilities (i.e. $f(Z; \theta) \in [0, 1]^{N \times K} \equiv P(S|Z, \theta)$) and $P(Z)$ is a generative model for the latent phenotypes themselves. Additional details regarding $f(Z; \theta)$ and $P(Z)$ are provided in the Supplementary Methods.

Given an observed symptom matrix (denoted $S = s$), we obtained estimates for the symptom risk function parameters (denoted $\hat{\theta}$) by optimizing a lower bound approximation to the model marginal likelihood (i.e. $P(s|\theta) = \int P(s, Z|\theta)dZ$) using an amortized, variational inference algorithm[27,28]. Model inference was conducted using the training subsets only. Estimates for the latent phenotypes of interest (denoted $\hat{Z}$) were obtained as a direct by-product of this optimization process (see Supplementary Methods). In practice, the observed symptom matrices for each rare disease were constructed from the UCSF-HPO, the UKBB-HPO, and the UCSF-HPO-UKBB datasets using the annotations available on the HPO website (see Supplementary Data 3 for the complete disease-to-symptom mappings). However, some of the aligned symptoms were manually curated to resolve convergence issues (see Supplementary Methods); Supplementary Data 5 contains the final disease-to-symptom mappings used to infer the cryptic phenotypes for the 10 diseases that passed all our filters (see below). Additional details concerning our model inference and evaluation procedures are provided in the Supplementary Methods.

*Cryptic phenotype identification and evaluation.* Following inference, we assigned each rare disease a single cryptic phenotype, which we define as the latent trait that best captures the symptom frequency intrinsic to the rare disease of interest (i.e. its morbidity). By default, all our models were initialized with a total of 10 possible latent phenotype components, as multiple pathologic processes can contribute to the correlation structure observed among some set of symptoms (see Supplementary Methods for more information). Although this meant that many of our models were initially overdetermined, we found that our inference algorithm was able to automatically remove unnecessary components by zeroing out their parameters in the symptom risk function. The number of latent components that remained following model inference was termed the model's effective rank ($L_{eff}$, see Supplementary Methods for precise definition), which was typically much less than the number of components used to initialize the model (Supplementary Fig. 4). When $L_{eff} = 1$, then this single component was automatically selected to

be the disease's cryptic phenotype. When $L_{eff} > 1$, then each inferred latent phenotype was used separately as a classifier to predict rare disease diagnoses in the training dataset, noting that the component that best captures the morbidity of a disease should be most predictive of its diagnostic status (see Fig. 3a–c, inset for examples). This top-performing latent component (assessed using the average precision score implemented in scikit-learn[70]) was then selected as the disease's cryptic phenotype.

Note, the model fitting described above was completed in both the UCSF and UKBB datasets, with the caveat that not all the Mendelian diseases in Supplementary Data 1 map to specific ICD10 diagnostic codes in the UKBB dataset (the encoding for this dataset is more limited, see above). Therefore, all cryptic phenotype models inferred in the UKBB dataset were also applied to the UCSF dataset (using UCSF-HPO-UKBB, see Fig. 2e for results). To ensure the assigned cryptic phenotypes were in fact capturing Mendelian-disease related morbidity, we compared the average cryptic phenotype severity among diagnosed cases to their undiagnosed controls (using the test datasets only). For a cryptic phenotype to successfully capture disease morbidity, the average symptom severity among Mendelian disease cases had to be significantly higher in both the UCSF and UKBB datasets (significance assessed through bootstrapped re-sampling[71] after performing Bonferroni corrections, see Fig. 2d and e for results). If Mendelian disease diagnostic codes were not available in the UKBB, then this increase in cryptic phenotype severity only needed to occur in the UCSF dataset.

Beyond capture, we also wanted to ensure that models inferred within the two independent datasets were consistent, meaning that they generated similar results when applied to the same dataset. Therefore, the phenotype models inferred within the UKBB were directly applied to the UCSF-HPO-UKBB dataset. Consistency was then assessed in three ways. First, the same latent component had to be assigned as the cryptic phenotype in both datasets (see above). Second, the UKBB model had to reproduce the increase in phenotype severity observed among the Mendelian disease cases within this new dataset. Third, the cryptic phenotypes produced by the UCSF and UKBB models needed to be correlated (as assessed through the coefficient of determination, $r^2$). Using an $r^2$ cutoff of 0.2, ten of the original fifty Mendelian disorders survived our capture, replication, and consistency filters. However, it is entirely plausible that replicable and consistent cryptic phenotypes could have been inferred for the other disorders through careful curation of annotated symptoms, larger sample sizes, and more focused adjustment of inference algorithm parameters (see Supplementary Methods).

**Cryptic phenotype validation.** The cryptic phenotypes for the five diseases listed in Table 1 were further validated through rare variant association studies. This required identifying pathogenic variant carriers within the UKBB. For A1ATD, the causal Pi*Z allele (rs28929474) was directly ascertained through array-based genotyping, so carriers of the Pi*MZ and Pi*ZZ genotypes were identified in the call/imputation files (see UKBB Data Category 263). For the remaining diseases, we downloaded the VCF files that contained the known causal genes (see Table 1). We then used the ClinVar database VCF (available at https://ftp.ncbi.nlm.nih.gov/pub/clinvar/) to identify all variants in the UKBB that have pathogenic/likely pathogenic annotations (accomplished using bcftools[72]). Because heterozygous loss-of-function (LoF) is an established molecular mechanism for each of diseases in Table 1 (except for A1ATD), we also identified LoF variants that were not listed in ClinVar. These were annotated using the LOFTEE plugin[29] for the ensembl variant effect predictor[73]. Not all the variants isolated in this manner have equivalent levels of evidence for pathogenicity. Therefore, we added a flag to each variant to indicate if: (1) it had conflicting annotations, (2) it was annotated by a single submitter, or (3) it was located within a non-canonical transcript (LoF variants only). Supplementary Data 6 contains a complete list of the P/LP variants analyzed in this study.

Using the VCF and genotype call files, we then identified all carriers of the P/LP variants described above. To assess whether the variants were associated with cryptic phenotype severity, we estimated their average genetic effect using the following linear model:

$$CP_i = \beta_0 + \beta_{P/LP} \times G_i + \boldsymbol{a} \times \boldsymbol{X}^T \quad (2)$$

where $CP_i$ denotes the cryptic phenotype of the $i$th subject, $\beta_0$ is an intercept parameter, $\beta_{P/LP}$ is the average effect parameter for the P/LP variants, $G_i$ is the carrier status of the $i$th patient, and $\boldsymbol{X}^T$ denotes a vector of covariates (with their corresponding parameter vector given by $\boldsymbol{a}$). Sex, age (inverse rank-transformed to remove skew), UKBB array platform, and the first 10 principal components of the genetic relatedness matrix were used as covariates. The analysis was limited to unrelated individuals of similar ancestry (Caucasian) to reduce the risk for population structure confounding ($N = 153,182$). Estimates for the parameters were produced using ordinary least squares, and per-parameter significance was assessed using a two-sided $T$-test. To account for effects related to variant annotation, we also fit the following linear model:

$$CP_i = \beta_0 + \beta_{P/LP} \times G_i + \beta_{Unflagged\ P/LP} \times G_i \times U_i + \boldsymbol{a} \times \boldsymbol{X}^T \quad (3)$$

where $U_i$ is a binary variable that indicates if the $i$th patient carries a variant without any annotation flags (see above). This enabled us to decompose the phenotypic contributions of P/LP variants into baseline ($\beta_{P/LP}$) and unflagged ($\beta_{Unflagged\ P/LP}$) effects, which are displayed at the top of the panels in Fig. 3a–c and

Supplementary Fig. 6b. Note, the AS phenotype is known to be more severe among hemizygous male carriers of COL4A5 pathogenic variants, consistent with X-linked inheritance. Therefore, we included an interaction term between sex and COL4A5 carrier status during our molecular validation of the AS cryptic phenotype. This interaction effect did not reach statistical significance ($\beta_{COL4A5xSex} = 0.09 \pm 0.15$; $P$-value = 0.56), likely due to the small number of male P/LP COL4A5 carriers in the dataset ($N = 12$ in UKBB-Unrelated). As a result, sex-specific interaction effects were not included in downstream analyses.

**Common variant genome-wide association analyses.** Genome-wide association studies were performed to identify common genetic variants associated with the cryptic phenotypes assigned to the diseases in Table 1. To reduce the risk of confounding, the association analyses were conducted using a subset of patients isolated from UKBB-Unrelated ($N = 342,796$) that met the following criteria: (1) did not possess a P/LP variant in a gene linked to the disease of interest, (2) were never diagnosed with this disease, and (3) were not a 3rd degree or closer relative of any of these subjects. From this training cohort, a random subset of 10% were removed and added to the Mendelian disease P/LP carriers. This second dataset is called the target cohort, and it was used to perform polygenic score replication and validation. All SNPs meeting the following criteria were in included into the analyses: directly genotyped by the UKBB, minor allele frequency (MAF) ≥ 1%, missing genotype fraction ≤5%, and Hardy–Weinberg equilibrium (HWE) $P$-value ≥ $10^{-12}$. Note, a relatively limited number of genetic markers (579,429 SNPs) met these criteria, but this smaller set of features enabled us construct individual-level prediction models for polygenic score inference (see below).

Genome-wide association studies (GWAS) were conducted by fitting the following linear model to each cryptic phenotype:

$$CP_i = \beta_0 + \beta_j^{SNP} \times G_{i,j} + \boldsymbol{a} \times \boldsymbol{X}^T \quad (4)$$

where $CP_i$ indicates the cryptic phenotype in the $i$th patient, $\beta_j^{SNP}$ represents the average effect of the $j$th SNP, $G_{i,j}$ encodes the minor allele count ($G_i \in \{0, 1, 2\}$; additive model), and $\boldsymbol{X}^T / \boldsymbol{a}$ denote covariates/effect parameters respectively. Sex, age (rank-normalized), UKBB array platform, and the first 10 principal components of the genetic relatedness matrix were used as covariates. Association statistics were estimated using the Plink2[74] software package (--glm command). Lead SNPs and their corresponding annotations were obtained using the FUMA[75] platform. The loci identified for the three diseases with genome-wide significant effects are provided as Supplementary Tables 2–4 (A1ATD, AS, and ADPKD, respectively).

SumHer (available within the LDAK toolkit[76]) was used to produce estimates for the fraction of the additive variance explained by the genotyped SNPs (narrow-sense heritability, denoted $h^2$). This required the specification of an underlying heritability model[76]. Based on recommended best-practices, we used the LDAK-Thin model given its simplicity and portability to individual-level prediction. This required computing a tagging file, which was constructed using a random subset ($N = 10,000$) of UKBB-Unrelated. First, duplicate SNPs were identified using the ldak-thin command with the following options:--window-prune .98--window-kb 100. Next, the tagging file itself was constructed using the ldak --calc-tagging command (with options --power -.25--window-cm 1 --save-matrix YES). Finally, narrow-sense heritability estimates were produced from the GWAS summary statistics using the ldak --sum-hers command (while also storing the per-SNP heritability estimates for downstream analyses).

Polygenic prediction models summarizing the common variant association statistics were inferred for the three diseases in Table 1 that had cryptic phenotype $h^2$ estimates significantly >0 (A1ATD, AS, and ADPKD). These models were estimated using the individual-level genotype data available for each training cohort. More specifically, we used LDAK-Bolt-Predict[32] (ldak --bolt command) to estimate effect sizes for every SNP included in the cryptic phenotype association analyses (while conditioning on the covariates included in the initial linear model, see above). This required access to the per-SNP heritability estimates, which were produced by SumHer (see above). Note, 10% of the training data was withheld during model inference (using the --cv-proportion .1 flag) to estimate prior parameters. After model fitting was complete, polygenic scores were imputed into the target cohort using the --calc-scores command (with --power flag set to 0). The per-SNP effect size estimates produced by the predictor models are included as Supplementary Data 7–9 (A1ATD, AS, and ADPKD respectively).

**Estimating the effects of polygenic load on Mendelian disease severity and outcomes.** Polygenic scores (PGS) were imputed into the target cohorts for each rare disease in order to: (1) replicate the PGS-cryptic phenotype relationships, (2) assess for interaction effects between the PGS and P/LP variants, and (3) determine if high polygenic load was associated with established Mendelian disease outcomes.

The first two analyses were accomplished by fitting the following linear model within the target cohort constructed for each cryptic phenotype:

$$CP_i = \beta_0 + \beta_{PGS} \times \xi_i + \beta_{P\backslash LP} \times G_i + \beta_{PGSxP\backslash LP} \times G_i \times \xi_i + \boldsymbol{a} \times \boldsymbol{X}^T \quad (5)$$

where $\xi_i$ represents the PGS for the $i$th patient, $\beta_{PGS}$ represents its average phenotypic effect, and $\beta_{PGSxP\backslash LP} \times G_i \times \xi_i$ models the interaction between the PGS and the P/LP variants. In the case of A1ATD, the two pathogenic genotypes (Pi*ZZ

and Pi*MZ) were modeled as separate genetic effects, each with their own PGS interaction terms. For AS, both flagged and unflagged P/LP variants were included into the analysis, as they were both shown to influence cryptic phenotype severity (see Supplementary Fig. 6b). For ADPKD, only the unflagged variants were included, as there was no detectable phenotypic effect for the flagged variants, suggesting that they most likely represent annotation noise (see Fig. 3c). The previous model was fit using ordinary least squares, and association statistics were computed using two-sided $T$-tests.

Regarding covariates (i.e. $\mathbf{a} \times \mathbf{X}^{\mathrm{T}}$), sex, age, UKBB array platform, and the first 10 principal components of the genetic relatedness matrix were included in every model. Smoking history was also included into each model, although its incorporation varied across diseases. For A1ATD and AS, self-reported pack-years (defined by Data Field: 20161) were used to quantify smoking history. Note, there was a significant interaction effect between the Pi*Z allele and smoking history (as expected), so interaction terms between pack-years and the pathogenic genotypes were included into the regression model for this disease (see Fig. 5b). There was no significant interaction effect between pack-years and P/LP carrier status for AS ($\beta_{\mathrm{Pack\text{-}years\ x\ P/LP}} = 0.03 \pm 0.05$; $P$-value = 0.55), so smoking interaction terms were not included for this disorder.

Regarding ADPKD, smoking had a strong protective effect on cryptic phenotype severity such that those P/LP carriers with a history of ever-smoking (provided by Data Field: 20160) had systematically lower cryptic phenotype scores ($\beta_{\mathrm{Smoke\ x\ P/LP}} = -0.41 \pm 0.11$; $P$-value = $1.2 \times 10^{-4}$). This result is clearly at odds with the known pathophysiology of smoking and renal disease, and it likely stems from the fact that subjects with moderate-to-severe ADPKD are often diagnosed at a young age, prior to when smoking behavior is established (see Supplementary Fig. 9d for Kaplan–Meier curve of ESRD among P/LP carriers). Consistent with this hypothesis, significantly fewer P/LP carriers reported ever-smoking when compared to the general population (see Supplementary Fig. 9b). Based on these results, the relationship between smoking history and ADPKD severity is likely to be confounded by multiple unmeasured factors (specifically, medical intervention and counseling). Given our inability to adequately adjust for such complex confounding, smoking history in ADPKD was modeled using a simple binary variable (UKBB Data Field: 20160), which was included along with a P/LP interaction term. Note, similar confounding likely plays a role in the interaction effects between smoking and genotype for the other disorders (see Fig. 5e and Supplementary Fig. 7b for examples), but it was only significant enough to reverse the established morbidity relationship for ADPKD.

To confirm a role for polygenic load on Mendelian disease outcomes, we examined its effect on quantitative measurements that capture established pathophysiology but are distinct from the symptoms used to construct the cryptic phenotype. For A1ATD, we used the FEV1/FVC ratio (UKBB Data Field: 20258), a measurement derived from spirometry that quantifies the severity of obstructive lung disease (see Supplementary Fig. 7c). For AS, we examined urine microalbumin level (UKBB Data Field: 30500), which correlates with renal health and glomerular barrier function (see Supplementary Fig. 8c). Finally, for ADPKD, we computed an estimate[46] of the glomerular filtration rate (eGFR) from the serum creatinine level (UKBB Data Field: 30700), which is often used as a proxy for overall renal function (see Supplementary Fig. 9c). The regression models themselves incorporated the same genetic and covariate effects that were used for the cryptic phenotypes, and they were again fit using ordinary least squares with association statistics computed using two-sided $T$-tests.

Finally, the effect of polygenic load on Mendelian disease severity was assessed by estimating its association with: (1) the rare disease diagnosis itself (when available) and (2) the onset of clinically important outcomes. The effect of the PGS on Mendelian disease diagnostic risk was modeled using logistic regression according to

$$\text{Log-Odds}(D_i) = \beta_0 + \beta_{\mathrm{PGS}} \times \xi_i + \beta_{\mathrm{P\backslash LP}} \times G_i + \beta_{\mathrm{PGSxP\backslash LP}} \times G_i \times \xi_i + \mathbf{a} \times \mathbf{X}^{\mathrm{T}} \quad (6)$$

where $D_i$ is in a binary variable indicating whether a disease diagnosis is present or absent. The covariates included were sex, age, UKBB array platform, the first 10 principal components of the genetic relatedness matrix, and smoking history (plus interaction terms where relevant, see above). Model fitting was performed using the maximum-likelihood method with a Firth penalty term, which was included given the risk for Type I error rate inflation in the setting of unbalanced samples and rare predictors[77]. Significance for a given association was assessed using a likelihood-ratio $\chi^2$ test[78].

The age-of-onset for clinically important Mendelian disease outcomes was also used to assess the effects of polygenic load on disease severity. The outcomes included in this study were: end-stage renal disease (ESRD; UKBB Data Field: 42026), chronic obstructive pulmonary disease (COPD; UKBB Data Field: 42016), recurrent and persistent hematuria (UKBB Data Field: 132002), and cystic kidney disease (UKBB Data Field: 132532). Details concerning the construction of these data fields are available through the UKBB. For each outcome, age-of-onset was modeled using Cox proportional hazards (CPH) regression:

$$\lambda_i = \beta_{\mathrm{PGS}} \times \xi_i + \beta_{\mathrm{P\backslash LP}} \times G_i + \beta_{\mathrm{PGSxP\backslash LP}} \times G_i \times \xi_i + \mathbf{a} \times \mathbf{X}^{\mathrm{T}} \quad (7)$$

where $\lambda_i$ represents the logarithm of the partial hazard function for the $i$th subject. The following covariates were included into the model: sex, UKBB array platform, the first 10 principal components of the genetic relatedness matrix, and smoking history (with interaction terms as described above). Model fitting was performed by

maximizing the partial likelihood (using the `lifelines` software package[79]), and significance was assessed using a likelihood-ratio $\chi^2$ test.

**Reporting summary.** Further information on research design is available in the Nature Research Reporting Summary linked to this article.

## Data availability

The clinical and genetic datasets used in the analyses presented in this manuscript cannot be shared directly with third parties, as both have specific provisions against open data sharing outside of their usual application processes. Information regarding third party access to the UCSF De-Identified Clinical Data Warehouse can be found through UCSF Data Resources: https://data.ucsf.edu/cdrp/research, and the application process for access to the UK Biobank is outlined on their website: https://www.ukbiobank.ac.uk/register-apply. The ClinVar[2] (downloaded 24 March 2021), Human Disease Ontology[64] (downloaded 21 August 2019), and Human Phenotype Ontology[26] (downloaded 8 August 2019) databases are freely available online (https://www.ncbi.nlm.nih.gov/clinvar/, https://obofoundry.org/ontology/doid.html, and https://hpo.jax.org/app/, respectively). Additional databases used to align diseases and symptoms to structured medical data (along with their associated URLs for access) are provided in the Supplementary Methods (see Section 1.2.2). Datasets that were generated to conduct the analyses described in this manuscript are provided as Supplementary Data 1–9. The summary statistics for the cryptic phenotype genome-wide association studies were submitted to the GWAS catalog[36] at https://www.ebi.ac.uk/gwas/ (accession IDs: GCST90101825, GCST90101826, GCST90101827, GCST90101828, and GCST90101829 for A1ATD, HHT, MFS, AS, and ADPKD, respectively). The cryptic phenotypes for A1ATD, HHT, MFS, AS, and ADPKD were returned to the UK Biobank for third-party use (under Application ID 53312).

## Code availability

We have deposited the software developed for this study. Latent phenotype model inference was performed using the vLPI software package, which was specifically designed to perform the analyses presented in this manuscript. It is written in Python and relies heavily upon the Pyro[80] (version 1.3.1) and PyTorch[81] (version 1.5.1) software libraries. The vLPI software package is available via Github[59]. To facilitate replication, a software package that automatically imputes the cryptic phenotypes analyzed in this study into new structured clinical datasets (CrypticPhenoImpute) is available via Github[82]. Finally, the scripts used to perform the analyses described in this manuscript are also available on Github[83]. A script for building a singularity container with the vLPI software package installed is available as well: https://github.com/daverblair/singularity_vlpi.

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

## Acknowledgements

This work was supported by the Stimulating Access to Research in Residency (StARR) program at UCSF (NHLBI grant 5R38HL143581-03; PIs Alison Huang and Kirsten Bibbins-Domingo), which provided funding for the first author and data access. The first author was also supported by the UCSF Pediatrics and Medical Genetics Residency Programs. We thank members of the Shieh lab for their feedback given throughout. This research has been conducted using the UK Biobank Resource under Application Number 53312; we are extremely grateful to the organization and its participants for providing access to this data. We also want to thank the Academic Research Systems at UCSF for providing access to the UCSF Clinical Data Warehouse, and the Wynton High Performance Computing team for their maintenance of the computational resources used in this study. This study was reviewed by the UCSF Institutional Review Board (IRB #: 19-29458) and qualified for Exempt status.

## Author contributions

D.R.B. and J.T.S. conceived of the project. D.R.B. designed the study, implemented the analyses, interpreted the results, made the figures, and wrote the manuscript with input from T.J.H. and J.T.S.

## Competing interests

The authors declare no competing interests.
