## [Peer Review File · Nature Communications]

Common genetic variation associated with Mendelian disease severity revealed through cryptic phenotype analysisEditorial Note: This manuscript has been previously reviewed at another journal that is not operating a transparent peer review scheme. This document only contains reviewer comments and rebuttal letters for versions considered at *Nature Communications*.

REVIEWER COMMENTS

Reviewer #1 (Remarks to the Author):

Building upon a probabilistic generative model, this paper proposes cryptic phenotype analysis to generate pseudo continuous traits from symptom data for disease variability/severity. In other words, in contrast to clinical setting where symptoms are measured to decide whether a disease exists (Yes/No), this paper mines the existing symptoms in electronic medical records to infer a score for the disease. They established the relevance of the inferred score with the appearance and/or severity of Mendelian disease by considering associations with known disease variants from exome sequencing. They applied the validated traits for GWAS and polygenic score analysis to identify common variants associated with the inferred traits. This approach identified a few effect modifiers (smoking status and polygenic scores) for A1ATD, Alport Syndrome and Autosomal-Dominant Polycystic Kidney Disease. With considerable revision, the flow of this manuscript is clear and the statistical concerns have been addressed. The language is a bit hard to follow, and I would suggest a significant English editing for publication.

Minor: Genome inflation factor increases with sample size. It would be good to report a standardized version.

Reviewer #3 (Remarks to the Author):

This study is an interesting study that aims to develop a challenging approach to understanding clinical phenotype heterogeneity within patients that harbor rare-disease variants. It has been observed that even in families with the same pathogenic clinical variant, the expressivity and penetrance of disease can vary drastically. This is believed to be at least partially due to clinical variants that focus on identification of the clinically detected spectrum of phenotypes within rare disease.

The early reviewers noted several criticisms that in this new version, I believe to be mostly addressed. The major revisions here include additional disease phenotypes are explored in response to the major criticism that only a 1-2 phenotypes were explored in the original version. This new version includes 5 traits (out of an initial 50) where they deeply explored CPA. I think it may still be unclear how universal this method would be across rare disease phenotypes and what factors (age at diagnosis, dx criteria, rarity of phenotype, etc) make certain diseases more amenable to use of this approach.

This type of research is not easy and ultimately limited by the datasets that are existence, which have their own challenges. There are always going to be biases in biobank-related data where participants are not uniformly tested for every disorder, phenotype or endophenotype. This can make backing out latent and cryptic phenotypes that make up the full spectrum of disease challenging, noisy and imprecise.

The major findings here are that in some types of rare disease, you can take phenotype data and back out cryptic phenotypic signals that represent a clinical severity spectrum from EHR ICD10 data. Here they tested their hypothesis on 50 disorders and found of these 10 were 1) transferable between UCSF and UKBBB phenotypes and 2) for these 10, 3 of them had a GWAS performed were significant deviation

from expected demonstrating that common genetic variants (perhaps along with environment) play a significant role in modulating phenotype expression. This idea, that genetic background can alter phenotype expression and disease severity, is something that we in the genetics community have thought a lot about and observed in other model systems. However, demonstration in humans has been incredibly challenging due to sample sizes and lack of well established phenotype data. This study, which aims to systematically identify milder phenotypes within the spectrum of rare and typically severe genetic disorders is a novel approach that , especially upon this revision has included additional controls.

While there remain many improvements that could taken, where only 3 out of 50 phenotypes that were originally tested can conclusively demonstrate that there is a genetic background effect that can be identified by stratifying phenotypes using the cryptic phenotype analysis is an interesting finding and warrants further development and assessment as a tool for biobanks leveraging EHR data. I do think that this is the important first step in demonstrating that critical importance of working with the phenotypes we have. Based on my reading of the reviews, the authors have addressed the concerns of the initial round of reviewers.

Major Comments

One major question is what are the limitations of this method for identifying cryptic phenotypes within the EMR. It would appear that both the rarity of the condition, as well as the number of clinical tests that can be used as part of the diagnostic validation would alter the ability to identify these latent phenotypes. I might suggest that although it is compelling to bridge this study as linking Mendelian diseases with more common modifiers, it would appear that this might be most beneficial in complex traits (i.e. rheumatological disorders like lupus, as suggested by authors) to identify genomic regions associated with sub-phenotypes/cryptic phenotypes

This study focuses exclusively on structured data types, such as ICD-10 codes. However, much of the cryptic phenotypes might benefit from transformation of clinical notes (unstructured data) into structured data types. The authors might discuss the potential benefits of natural language processing of clinical notes and more unstructured clinical data into their phenotypes.

In addition to the larger biobanks linked with genomic data, such as UKBB, Japan Biobank, and Regeneron Data, there are a number of US biobanks linked with genetic datasets of diverse ancestry that are coming online. These include the biobanks at Mt Sinai, UCLA, Colorado, Vanderbilt and other institutions.

Minor Comments

Page 3. "For conditions like familial hypercholesterolemia¹⁸, hereditary breast cancer¹⁹, and long QT syndrome²⁰, this relationship is well documented. As a result, the interplay between rare and common genetic variation has been systematically investigated ^{21–24}."

I'm not sure that this statement is true. These disorders has not been systematically investigated. For example in ref 18 for FH, they show that common variant load can result in something that is equivalent to a monogenic mutation causing FH-- but the integration of the two is not clearly defined. A better reference for this point might be : <https://doi.org/10.1038/s10038-021-00929-7>. Although the

conclusion here is that PRS doesn't change much once a monogenic mutation is identified and that PRS is higher when patients are mutation negative.

“Critically, this spectrum of variation cannot be measured directly. Instead, the trait is analyzed implicitly by a clinician, who translates their observations into a set of symptoms (Figure 1A, upper left).”

It seems like it should be lower left, not upper left in this figure?

REVIEWER COMMENTS

Reviewer #1 (Remarks to the Author):

Building upon a probabilistic generative model, this paper proposes cryptic phenotype analysis to generate pseudo continuous traits from symptom data for disease variability/severity. In other words, in contrast to clinical setting where symptoms are measured to decide whether a disease exists (Yes/No), this paper mines the existing symptoms in electronic medical records to infer a score for the disease. They established the relevance of the inferred score with the appearance and/or severity of Mendelian disease by considering associations with known disease variants from exome sequencing. They applied the validated traits for GWAS and polygenic score analysis to identify common variants associated with the inferred traits. This approach identified a few effect modifiers (smoking status and polygenic scores) for A1ATD, Alport Syndrome and Autosomal-Dominant Polycystic Kidney Disease. With considerable revision, the flow of this manuscript is clear and the statistical concerns have been addressed. The language is a bit hard to follow, and I would suggest a significant English editing for publication.

We appreciate the reviewer's comments and have revised the text to improve clarity.

Minor: Genome inflation factor increases with sample size. It would be good to report a standardized version.

This is now addressed in the main text. In addition, Supplementary Table 2 was added to the manuscript, which provides the genomic inflation factors adjusted for sample size.

Reviewer #3 (Remarks to the Author):

This study is an interesting study that aims to develop a challenging approach to understanding clinical phenotype heterogeneity within patients that harbor rare-disease variants. It has been observed that even in families with the same pathogenic clinical variant, the expressivity and penetrance of disease can vary drastically. This is believed to be at least partially due to clinical variants that focus on identification of the clinically detected spectrum of phenotypes within rare disease.

The early reviewers noted several criticisms that in this new version, I believe to be mostly addressed. The major revisions here include additional disease phenotypes are explored in response to the major criticism that only a 1-2 phenotypes were explored in the original version. This new version includes 5 traits (out of an initial 50) where they deeply explored CPA. I think it may still be unclear how universal this method would be across rare disease phenotypes and what factors (age at diagnosis, dx criteria, rarity of phenotype, etc) make certain diseases more amenable to use of this approach.

We understand that the universality of the methods/assumptions is somewhat uncertain, and we now address this issue in the revised Discussion. Systematically validating its underlying assumptions across a wide range of diseases, including other Mendelian and even complex disorders, would require substantial analyses. This is beyond the scope of the current manuscript, which sought to test the idea of using cryptic quantitative traits to summarize the high-dimensional phenotypes associated with Mendelian diseases. Future work will focus on applying the approach to a wider variety of diseases while also improving the methods used to infer the cryptic phenotypes.

This type of research is not easy and ultimately limited by the datasets that are existence, which have their own challenges. There are always going to be biases in biobank-related data where participants are not uniformly tested for every disorder, phenotype or endophenotype. This can make backing out latent and cryptic phenotypes that make up the full spectrum of disease challenging, noisy and imprecise.

We completely agree. This is why we replicated our inferred phenotypes in two independent datasets (Figure 2). Moving forward, methods will be needed that can robustly infer, replicate, and transfer these cryptic phenotypes across datasets. This will be the focus of future work and is addressed in the revised Discussion section.

The major findings here are that in some types of rare disease, you can take phenotype data and back out cryptic phenotypic signals that represent a clinical severity spectrum from EHR ICD10 data. Here they tested their hypothesis on 50 disorders and found of these 10 were 1) transferable between UCSF and UKBBB phenotypes and 2) for these 10, 3 of them had a GWAS performed were significant deviation from expected demonstrating that common genetic variants (perhaps along with environment) play a significant role in modulating phenotype expression. This idea, that genetic background can alter phenotype expression and disease severity, is something that we in the genetics community have thought a lot about and observed in other model systems. However, demonstration in humans has been incredibly challenging due to sample sizes and lack of well established phenotype data. This study, which aims to systematically identify milder phenotypes within the spectrum of rare and typically severe genetic disorders is a novel approach that , especially upon this revision has included additional controls.

While there remain many improvements that could taken, where only 3 out of 50 phenotypes that were originally tested can conclusively demonstrate that there is a genetic background effect that can be identified by stratifying phenotypes using the cryptic phenotype analysis is an interesting finding and warrants further development and assessment as a tool for biobanks leveraging EHR data. I do think that this is the important first step in demonstrating that critical importance of working with the phenotypes we have. Based on my reading of the reviews, the authors have addressed the concerns of the initial round of reviewers.

We agree that there are multiple avenues available for further investigation, which are discussed in the revised manuscript (see below).

Major Comments

One major question is what are the limitations of this method for identifying cryptic phenotypes within the EMR. It would appear that both the rarity of the condition, as well as the number of clinical tests that can be used as part of the diagnostic validation would alter the ability to identify these latent phenotypes. I might suggest that although it is compelling to bridge this study as linking Mendelian diseases with more common modifiers, it would appear that this might be most beneficial in complex traits (i.e. rheumatological disorders like lupus, as suggested by authors) to identify genomic regions associated with sub-phenotypes/cryptic phenotypes.

Yes. There are many future applications for the ideas and methods described in this study. We have since revised the Discussion section to make this point, and there is now a paragraph addressing how the approach might be useful for the investigation of complex diseases.

This study focuses exclusively on structured data types, such as ICD-10 codes. However, much of the cryptic phenotypes might benefit from transformation of clinical notes (unstructured data) into structured data types. The authors might discuss the potential benefits of natural language processing of clinical notes and more unstructured clinical data into their phenotypes.

We agree that augmenting Cryptic Phenotype Analysis with additional structured and unstructured datatypes will likely be useful, and we intend to pursue this in future work. The revised Discussion section now specifically addresses this point. We agree that natural language processing could also help in transforming unstructured data for future analyses.

In addition to the larger biobanks linked with genomic data, such as UKBBB, Japan Biobank, and Regeneron Data, there are a number of US biobanks linked with genetic datasets of diverse ancestry that are coming online. These include the biobanks at Mt Sinai, UCLA, Colorado, Vanderbilt and other institutions.

We agree and would like to apply Cryptic Phenotype Analysis to other datasets as they become more widely available. As biobank-scale analyses become more common, robust methods for summarizing high-dimensional clinical phenotypes will be needed. CPA is an early step in this direction, and we intend to build upon the approach in future work. This is now specifically discussed in the revised Discussion section.

Minor Comments

Page 3. "For conditions like familial hypercholesterolemia¹⁸, hereditary breast cancer¹⁹, and long QT syndrome²⁰, this relationship is well documented. As a result, the interplay between rare and common genetic variation has been systematically investigated 21–24."

I'm not sure that this statement is true. These disorders has not been systematically investigated. For example in ref 18 for FH, they show that common variant load can result in something that is equivalent to a monogenic mutation causing FH-- but the integration of the two is not clearly defined. A better reference for this point might be : <https://doi.org/10.1038/s10038-021-00929-7>. Although the conclusion here is that PRS doesn't change much once a monogenic mutation is identified and that PRS is higher when patients are mutation negative.

This is a good point. We have since revised this sentence in the new version of the manuscript and included new references, including the one provided by the reviewer. Thank you.

"Critically, this spectrum of variation cannot be measured directly. Instead, the trait is analyzed implicitly by a clinician, who translates their observations into a set of symptoms (Figure 1A, upper left)."

It seems like it should be lower left, not upper left in this figure?

Agree, this oversight is corrected in the revised manuscript.